# Breast Cancer Detection in Mammogram Images Using K–Means++ Clustering Based on Cuckoo Search Optimization

**DOI:** 10.3390/diagnostics12123088

**Published:** 2022-12-07

**Authors:** Kittipol Wisaeng

**Affiliations:** Technology and Business Information System Unit, Mahasarakham Business School, Mahasarakham University, Mahasarakham 44150, Thailand; kittipol.w@acc.msu.ac.th; Tel.: +66-086-6393870

**Keywords:** breast cancer, mammogram images, K–means++ clustering, cuckoo search optimization

## Abstract

Traditional breast cancer detection algorithms require manual extraction of features from mammogram images and professional medical knowledge. Still, the quality of mammogram images hampers this and extracting high–quality features, which can result in very long processing times. Therefore, this paper proposes a new K–means++ clustering based on Cuckoo Search Optimization (KM++CSO) for breast cancer detection. The pre-processing method is used to improve the proposed KM++CSO method more segmentation efficiently. Furthermore, the interpretability is further enhanced using mathematical morphology and OTSU’s threshold. To this end, we tested the effectiveness of the KM++CSO methods on the mammogram image analysis society of the Mini–Mammographic Image Analysis Society (Mini–MIAS), the Digital Database for Screening Mammography (DDSM), and the Breast Cancer Digital Repository (BCDR) dataset through cross-validation. We maximize the accuracy and Jaccard index score, which is a measure that indicates the similarity between detected cancer and their corresponding reference cancer regions. The experimental results showed that the detection method obtained an accuracy of 96.42% (Mini–MIAS), 95.49% (DDSM), and 96.92% (BCDR). On overage, the KM++CSO method obtained 96.27% accuracy for three publicly available datasets. In addition, the detection results provided the 91.05% Jaccard index score.

## 1. Introduction

Breast cancer is a challenging and fatal disease that seriously affects women, leading to sudden death in critical cases. Based on the statistics provided by the World Health Organization (WHO) in 2020, there were 2.3 million women diagnosed with breast cancer [1]. As of the end of 2020, WHO reported that 685,000 deaths among women globally were caused by breast cancer. Automated detection of breast cancer from mammogram images is a controversial issue among researchers. Several image processing methods have been proposed. These include a variety of machine learning methods, such as Support Vector Machine (SVM), Convolution Neural Networks (CNN), Naïve Bayesian (NB), and Artificial Neural Networks (ANN). For instance, Karabatak et al. [2] proposed a new algorithm to detect breast cancer using NB on the Wisconsin breast cancer dataset. Additionally, Pak et al. [3] used non–subsampled contour transforms and super-resolution for breast cancer detection. Sabu et al. [4] attempted a comparison of ANN and SVM by selecting the best detection method among 60 cancerous and 60 non–cancerous images. This study focused on breast cancer tumor types by Mini–MIAS, and DDSM. Then, Progressive Support–pixel Correlation Statistical Method (PSCSM) was comparatively used by Yang et al. [5]. Global and local thresholding based on image histogram, Region Growing Method (RGM), Markov Random Field (MRF), stochastic relaxation methods, and fuzzy use membership function were used for breast cancer detection by Punitha et al. [6]. Al-masni et al. [7] suggested deep regional learning based on CNN. Shen et al. [8] proposed the Mixed-Supervision Guided Deep Model (MSGDM) based on U-net model and ResCU-Net. Different machine learning, including K–Nearest Neighbourhood (KNN), Fuzzy Clustering (FC), Neural Network (NN), Logistic Regression (LG), SVM, and Random Forest (RF), were applied for breast cancer detection. A similarity of three unsupervised learning, including Fuzzy Rough Segmentation (FRS), Rough Segmentation (RS), and Fuzzy Segmentation (FS), was proposed by Punitha et al. [9]. Zebari et al. [10] offered a thresholding method and binary morphological segmentation for breast cancer. Vaka et al. [11] presented end–to–end testing for breast cancer detection using Deep Neural Network with Support Value (DNNSV). Azlan et al. [12] proposed Principal Component Analysis (PCA) and SVM for breast cancer detection. Next, George et al. [13] developed hybrid methods by combining the CNN and SVM on the Breast Cancer Histopathological Database (BreakHis). Alanazi et al. [14] performed a CNN to detect breast cancer disease and took mammogram images from the Kaggle dataset. The details of the existing up-to-date methods for breast cancer classification and detection are illustrated in Table 1.

From a comparative review of machine learning and performance parameters used for evaluating the detection method, each method is suitable for detecting breast cancer on different datasets. Nevertheless, the accuracy of each technique still varies for different datasets. This work aimed to assess the reliability of the diagnosis system for automatically detecting breast cancer in mammogram images. The automated breast cancer detection from mammogram images comprises a series of contributions, which are listed as follows: (1) We resolved the problem of poor quality and contrast enhancement of the original image of the different datasets by using color normalization and noise reduction stages, (2) A comprehensive application that used machine learning such as K-Means (KM), Fuzzy C-Means (FCM), Fuzzy K-Means (FKM), and KM++CSO to select a suitable robust method for the coarse segmentation stage, (3) A fused method for finely detecting breast cancer disease associated with region-based segmentation based on Cuckoo Search Optimization (CSO) is applied, (4) A new method was applied to three open-access datasets (Mini–MIAS, DDSM, and BCDR) with different camera settings, intensities, and additional noise, and (5) Validation of the proposed method by comparing it with Sensitivity (SEN), Specificity (SPEC), Accuracy (ACC), and Jaccard index score.

To achieve all these, we organized the study into four sections. Section 1 presents the background of breast cancer problems and the current literature reviews. It also focuses on segment limitations and how to use the proposed KM++CSO to solve these limitations. Section 2 presents the pre-processing method and the proposed KM++CSO framework phases. Section 3 describes the conducted results and comparisons with the literature. Finally, the conclusion and the analysis of comparison KM++CSO and others are presented in Section 4.

## 2. Materials and Methods

Since mammogram image is nowadays in digital format, it is possible to create a Computer Aided Diagnosis (CAD) system that automatically segments breast cancer disease. Nevertheless, automatic breast cancer detection using a CAD system requires the education of human screeners or experts for further validation. Therefore, new breast cancer detection is necessary for classifying breast cancer from the image background without unnecessary interference from other regions. Typically, the mammogram image includes noise or artifacts, which could adversely affect breast cancer detection. Sometimes the image is as high-intensity in the center of the image and low–intensity in a region in the posterior margin. This research proposed a fusion method to detect breast cancer from the low-intensity of image and the block diagram for automatic detection of breast cancer, as illustrated in Figure 1.

### 2.1. Dataset and Image Acquisition

A mammogram image used in this study is drawn from different datasets, some of which are from the Mini–MIAS [5,9,15,16,17,18,19] and DDSM [12], while others are from BCDR [10]. We used the same images to evaluate their detection methods [3,4,5,6,9,10,12]. This dataset from DDSM, Mini–MIAS, and BCDR comprises 2620, 323, and 123 images, respectively. Mini–MIAS contains 323 images divided into 207 healthy and 116 abnormal images. DDSM includes 1609 scanned film mammography studies divided into 695 healthy and 914 abnormal images. BCDR contains 18 normal and 105 abnormal images from 123 images. Before discussing the mammogram image, it may be helpful to describe the structure of the image; see Figure 2 [5,9,10,12,15,16,17,18,19]. Figure 2a is an image from Mini–MIAS, Figure 2b is an image from INbreast, and Figure 2c is an image from BCDR, respectively.

### 2.2. Image Color Normalization

The pre-processing method in many stages is image color normalization, and image analysis is no exception. One of the commonest is the one proposed by Gonzalez [24]. The author adopts a method based on the histogram specification to enhance the intensity of the gray level of the mammogram image. Bejnordi et al. [25], proposed a Whole–Slide Image Color Standardizer (WSMCS) to estimate the background and assumed that a background image usually had distributed color. However, the processing time of WSMCS is very high processing time. Janowczyk et al. [26] proposed stain normalization by using sparse autoencoders for the image to obtain the normalized image. However, the majority involve methods to scan the same image multiple numbers of times. Therefore, it may cause false detection and some of the information lost. In this study, the global histogram specification [27] helps adjust the histogram of the original image or histogram modifications to enhance the contrast and create a new image. For normalizing the values in the range [0, G], the probability density function of the input and output mammogram image that follows a specified shape $f_{z}(z)$ for z∈[0,1] is denoted by a subscript (s). If the input is performed to attain processed (output) intensity levels (s), the result would be an image that also has a uniform probability density function using Equation (1).
(1)$$s=G(z)=\int_{0}^{z}f_{z}(w)dw$$
where f_z_(w) is the probability density function of the severity levels in the image, G is the grayscale limit (or range), and the transformation function for equalizing the histogram. An adjustable transformation function is generated by histogram specification in the process of the original image with L gray–level values. The implementation of histogram specification and the transformation function T(r) are defined as Equations (2) and (3), respectively.
(2)$$s_{k}=T(r_{k})=\sum_{j=0}^{k}f_{R}\left(r_{j}\right)\;for\;k=0,\frac{1}{L-1},\frac{1}{L-1},\ldots ,1$$
(3)$$s_{k}=G(z_{k})=\sum_{i=0}^{k}f_{Z}\left(z_{i}\right)for\;k=0,\frac{1}{L-1},\frac{1}{L-1},\ldots ,1$$

The shape of the histogram specification is necessary to get the output image for some distinctive implementations. After the implementation of histogram specification, the normalized image is obtained. Here, the images in Figure 3a,b demonstrate the reference image, and its histogram while Figure 3c,d display the obtained image for grayscale after the normalization by histogram and its histogram after histogram specification of grayscale.

### 2.3. Image Denoising

The histogram specification applied in the previous section is helpful for the enhancement of image contrast. However, this step has also increased the intensity of the noise in the image. Therefore, the median filtering [28] was specially designed to remove the noise over-amplification in the image to some extent so that the image looks real. This method runs a window size of a median filter of 3 × 3 through the whole image. In a window size of the median filter, the targeted pixel is defined as per Equation (4).
(4)$$f\left(x,y\right)=\mathrm{median}\left[\sum_{i=-2}^{2}\times \sum_{i=-2}^{2}\mathrm{window}\left(x-i,y-j\right)\right]$$

Median filtering works well when there is one out–layer value in the window. This technique gives a good denoising result when the neighboring pixel values are very close.

### 2.4. Breast Cancer Detection

In this subsection, we chose a comprehensive application that used machine learning, such as KM, FCM, FKM, and KM++CSO, to select the most suitable algorithm for the coarse segmentation stage. Then, we combined these works with the clustering method coupled with mathematical morphology and OTSU’s threshold.

#### 2.4.1. KM Clustering Method

KM clustering [29] is the simplest of machine learning that solves the well–known clustering problem. Given a dataset ${\left\{x\right\}}_{1}^{n}\subset {ℜ}^{d}$ and a set of initial center locations ${\left\{c_{j}\right\}}_{1}^{k}\subset {ℜ}^{d},$ we merely alternate between the following two steps repeatedly (by assigning each point x_i_ to its nearest center and relocating each center c_j_ to the centroid of all points assigned to it), until center locations converge. Formally, the KM task is an optimization problem that partitions data into clusters while minimizing the sum of the squared distances between each data point and the cluster center it belongs to. Given a set of data points ${\left\{x\right\}}_{1}^{n}\subset {ℜ}^{d}$ and a set of initial center locations ${\left\{c_{j}\right\}}_{1}^{k}\subset {ℜ}^{d}$, we defined the minimization objective function J(x, c) as the sum of squared errors given in Equation (5).
(5)$$J(x,c)=\sum_{i=1}^{n}\underset{j}{argmin}{\left\|x_{i}-c_{j}\right\|}^{2}$$

There are n ways to partition a set of n points into k possibly empty clusters. If we insist on strictly non-empty clusters, then the number of ways to partition the set is given by the Stirling numbers of the second kind k, which we define below in Equation (6).
(6)$$\left\{\begin{array}{l} n\\ k \end{array}\right\}=\frac{1}{k!}\sum_{j=0}^{k}{\left(-1\right)}^{k-j}\left(\begin{array}{l} k\\ j \end{array}\right)j^{n}$$

The results segmented by using standard KM clustering to mammogram images are depicted in Figure 4. In this stage, k = 2, 3, 4, 5, and 6 run 110 iterations, which are used as the initial values for starting the detection. Figure 4b displaying the image detected using k = 3 is a “good” value and gives satisfactory results. Since KM is the simplest and most frequently unsupervised, we can implement it efficiently. Moreover, KM requires a few computations for comparing distance among data points and grouping clusters. However, KM has several limitations. First, the cluster that the method produces may be arbitrary and far from the optimal clustering. Second, the technique requires the user to specify k and the number of clusters and provides an initial guess of cluster locations. Finally, KM takes a long time to run this application.

#### 2.4.2. FCM Clustering

FCM has been widely used and developed for biomedical image segmentation. The FCM, presented by Bezdek et al. [30], describes the fuzzy segmentation for the pixels by calculating the membership values for fuzzy sets. The objective function minimizes the FCM for the fuzzy C–partition U and many cluster centers V defined as per Equation (7).
(7)$$Minimize\;J_{FCM}(U,V)=\sum_{k=1}^{N}\sum_{i=1}^{C}{\left(u_{ik}\right)}^{m}d^{2}\left(x_{k},v_{i}\right)$$
where $X=\{x_{1},x_{2},\ldots x_{3}\}\subseteq {ℜ}^{p},$ U is the fuzzy C–partition, C is the number of cluster centers (In this work, an experimental variation from two to six is used), m represents the weight exponentially, d^2^(x_k_, v_i_) is a measuring distance between pixel x_k_ and cluster center of v_i_ (usually the Euclidean distance), N is the number of the feature vector in an image, u_ik_ is the membership values for fuzzy sets of pixel k in cluster i, vi is the cluster center for subset i in feature space $(U,V)=\sum_{k=1}^{N}\sum_{i=1}^{C}{\left(u_{ik}\right)}^{{}^{m}}d^{2}$ and is an overall weight sum of generalized A–error. On solving the objective function (U and V), minimizing an objective function will be obtained so that the subject is subject to the constraint in Equation (8) is satisfying.
(8)$$\sum_{i=1}^{C}\;u_{ik}=1,\;\forall k\in \{1,\ldots ,N\}$$
with these observations, we can decompose of J_FCM_ be (u_ik_, c_k_). The necessary conditions for (u_ik_, c_k_) are calculated using Equation (9) and Equation (10), respectively.
(9)$$u_{ik}=\frac{1}{\sum_{j=1}^{C}{\left(\frac{d_{ik}}{d_{jk}}\right)}^{2/(m-1)}}$$
(10)$$v_{i}=\frac{\sum_{k=1}^{N}{\left(u_{ik}\right)}^{m}x_{k}}{\sum_{k=1}^{N}{\left(u_{ik}\right)}^{m}}$$

The algorithm of FCM is composed as seen in the following steps: (1) Fix the initialize the fuzzy membership matrix c, 2 c < N; Fix m, 1 < m < ∞; (2) Initialize step of the fuzzy C–partition U(0); (3) Assume distribution to be normal steps as b = 1, 2, …, N; (4) Calculate the c fuzzy cluster centers v_i_ as per Equation (10); (5) Update U^(k)^, U^(k+1)^ by calculate the new calculated u_ik_ as per Equation (9); (6) Compute the change in the membership values using an appropriate norm; if U^(k)^ − U^(k+1)^ < T, STOP; otherwise, set k = k + 1 and return to steps 4 and (7) A termination criterion is needed to stop the iterative process when the objective function is less than (T) and otherwise return to step 2.

The images segmented as test images for applying the FCM method with the various clusters are illustrated in Figure 5b–e. Figure 5b displays the three clusters designated using k = 3 (57 iterations) and the segmented image. Figure 5c shows the four clusters formed by selecting k = 4 (62 iterations). Figure 5d displays the five clusters formed by selecting k = 5 (100 iterations). Figure 5e depicts the six clusters formed by selecting k = 6. Figure 5f shows the seven clusters formed by selecting k = 7. This study concludes empirically that k = 4 is a “good” value. This value forms further advantages of simplifying the update equations and can therefore speed up computer implementations of the method.

#### 2.4.3. FKM Clustering

FKM clustering [31] is one method that provides a better result than KM for overlapping datasets. This method is utilized for the dataset in which the data points are between the centers. The technique works on the objective to minimize by as Equation (11).
(11)$$F(X,C)=\sum_{i=1}^{n}\sum_{j=1}^{k}u_{ij}^{m}{\left\|x_{i}-c_{j}\right\|}^{2}$$
where m is the fuzzifier and any real number greater than 1, u_ij_ the degree of membership in which data points belong to each cluster x_i_ to the cluster center c_j_ with the limitation that $u_{ij}\geq 0$, $\sum_{j=1}^{k}u_{ij}=1\;\forall i$ and ${\left\|x_{i}-c_{j}\right\|}^{2}$ is the distance between the data points to calculate x_i_ and c_j_, is an assign of the distance of the n data point to the closest cluster by its sum of squared distance from the cluster’s centroid. KM is a popular method that has been used in many segmentation domains. It is a good candidate for an extension to work with fuzzy feature vectors. The iterative approach of optimizing the objective function F(X, C) by updating the degree of participation of the data point x_i_ to the cluster centroids c_j_ and the cluster center c_j_ results within the clustering of the data is calculated based on Equations (12) and (13), respectively.
(12)$$u_{ij}=\sum_{j=1}^{k}{\left\|x_{i}-c_{j}\right\|}^{\frac{-2}{m-1}}/\sum_{j=1}^{k}\left({\left\|x_{i}-c_{j}\right\|}^{\frac{-2}{m-1}}\right)$$
(13)$$c_{j}=\sum_{i=1}^{n}u_{ij}^{m}\cdot x_{i}/\sum_{i=1}^{n}u_{ij}^{m}$$

As the value of m increases, the algorithm becomes fuzzy. The proposed FKM is tested on the image by setting the number of clusters to K = 2. The resultant images are displayed in Figure 6b. We compared these methods to those FKM clustering methods’ capability to provide “good” classification performance as the number of clusters increases. Figure 6c–f show the breast cancer classification performance as set in the number of clusters K = 3, 4, 5, and 6, respectively. Figure 6d shows “good” breast cancer classification performance with a number of clusters k = 4.

#### 2.4.4. Optimization Design with KM++CSO

KM++ is an algorithm to initialize the number of clusters k, which is given as an input to the KM method [32]. Since selecting the appropriate value of K prior is difficult, KM++ provides a method for determining the optimum value of K before proceeding to cluster the data. Impressively, this method offers theoretical guarantees about the output clustering quality. KM++ initialization method chooses the first center c1 selected uniformly at random from the dataset X, but each successive center c_j_ is chosen with probability by Equation (14).
(14)$$\frac{D^{2}(c_{j})}{\sum_{x\in X}D^{2}(x_{i})}$$

Here D(c_j_) denotes the minimum distance from a data pixel x to the closest point between c_j_ and all previously chosen centers. Then, the essential component of KM is presented as follows: (1) A starting with center c1, a point is chosen uniformly at random from the dataset X; (2) Take another center c_i1_, $c_{i1}=x\in X$ with probability $D^{2}(c_{j})/\sum_{x\in X}D^{2}(x_{i})$ where D^2^(c_j_) is the shortest distance between the data point x to the closest center which is chosen; (3) Repeat step 1 to 2 until the values of k center are chosen, and (4) Perform the KM using the k center as the initial center. In step 2, “D^2^ weighting” is the applied weighting. As a result of the segmentation error of KM, this seeding algorithm gives out considerable improvements. The clustering performed by KM++ does not provide any guarantees that it theoretically performs the best clustering. So, we can use an improved method with CSO [33] to find a better solution for initial centroids. The Cuckoo’s reproductive behavior inspired it as a particular species by laying its eggs in the nests of other host birds or species. We can summarize more details for KM++CSO in the pseudocode (Algorithm 1).
**Algorithm 1:** KM++CSO.(1)Select an appropriate value for K which is the number of clusters and obtain a number of data points.(2)Randomly initialize of K cluster center (c_1_, …, c_k_)(3)Keep repeating steps 4 and 5 until the algorithm convergence to a certain value or the centroids do not change(4)In each generation, all data points (x_i_) are now assigned the point to the cluster with the closest centroids (c_1_, …, c_k_)(5)Assign each data point to the closest cluster centroids j = 1, 2, …, k(6)End(7)A solution of the objective function (CSO); f(x), x = (x_1_, x_2_, …, x_k_)(8)Populate the initial population of n host nests xi, which is generated at random(9)While (t < Max Generation) or (Stopping criterion) do(10)For each loop Cuckoo i do(11)Get a Cuckoo randomly (named i) and replaced it by performing intensity values(12)Evaluate its quality/fitness F_i_(13)Choose randomly a nest from n (named j)(14)If (F_i_ > F_j_) minimization them(15)Replace j with i (generate a new solution)(16)End if(17)The number of host nests is fixed with a probability (Pa) of the eggs with low fitness being abandoned and rebuilt other nests somewhere(18)Keep the best nests with a higher fitness value(19)Rank the solution and find the best current one(20)Pass the current best solutions to the next generation(21)End while

We can summarize the KM++CSO: (1) Domain space: Initial centroids by CSO generated, and the used KM++ start with generated centroids. The result of clustering after classification is measured by a Peak–Signal–to–Noise Ratio (PSNR) between the input image and output image; (2) Improved optimization: the closet cluster centroid using CSO and KM++ finds the optimal initial image centroids by CSO, so the automatic detection is done using a clustering result and (3) An acquisition function: although the initial parameter selections take extra time, KM++CSO converges very fast for the coarse segmentation stage and lowers the computation time. KM++ determines a set of k clusters with parameters set to 2, 3, 4, 5, and 6 (100 iterations). Then the best solutions to optimize the results obtained by the Cuckoo search algorithm using the fraction Pa = 0.2 is used to compare the capability in segmentation performance as the number of clusters increases. The resultant images are displayed in Figure 7b–f, respectively. Figure 7d shows “good” segmentation performance in the number of clusters (k) is set to 4.

#### 2.4.5. Binary Image Detection

In this stage, binary image segmentation using the Gaussian OTSU’s threshold [34] is used. Given an image: P(i) is the normalized frequency of i, and is calculated according to Equation (15).
(15)$$P(i)\frac{number\left\{\left(r,c\right)|image\left(r,c\right)=i\right\}}{\left(R,C\right)}$$

Here r represents an index for rows, and c is a column of the image. R is the number of rows, and C is the columns of the image respectively. The value calculates the “good” of the threshold at value T* by minimizing the within-class variance (weighted) and the between-class variance, which is the sum of the weighted variances of the two classes. If this minimum is not unique, obtain T* by averaging the T value corresponding to the various minimized segments. This threshold is determined by minimizing within-class intensity variance at any threshold, where T is calculated by Equation (16).
(16)$$\sigma_{w}^{2}=w_{mb}\left(T\right)*\sigma_{mb}^{2}\left(T\right)+w_{bc}\left(T\right)*\sigma_{bc}^{2}\left(T\right)$$

Here w_mb_(T) and w_bc_(T) are the probability of the number of mammogram background and breast cancer pixels in each class for this threshold T and variance of the two classes T0 with a set of intensity levels from 0 to T, $\sigma_{mb}^{2}\left(T\right)$ and $\sigma_{bc}^{2}\left(T\right)$ represent the variance of color values and variance of class T1 with a set of intensity level from T + 1 to L, or T1 = {T, T+, …, L–, L} and $\sigma_{w}^{2}$ is the weighted sum of group variances. Based entirely on computations performed on the histogram (1–D) of the image, the normalized histogram of pixels that will be detected as mammogram background (w_mb_) and breast cancer (w_bc_) pixels, is given by Equation (17).
(17)$$w_{mb}\left(T\right)=\sum_{i=1}^{T}P\left(i\right),w_{bc}\left(T\right)=\sum_{i=T+1}^{T}P\left(i\right)$$

The probability of pixel intensity (P) is computed for each pixel value in two separated clusters (namely background and breast cancer pixels) using the cluster probability functions expressed as per Equation (18).
(18)$$\mu_{mb}=\sum_{i=1}^{T}P\left(i\right)/W_{mb},\;\mu_{bc}=\sum_{i=T+1}^{L}P\left(i\right)/W_{bc}$$
where L is the number of pixels in a grayscale image containing information between 1 and L {1, 2, 3, …, L} and $\sum_{i=T+1}^{L}P(i)$ is the sum of P(i), where values between i = T + 1 and L { i = T + 1, T + 2, = T + 3, …, L}. The class variance of the background and breast cancer pixels is given by Equations (19) and (20).
(19)$$\sigma_{mb}^{2}\left(T\right)=\frac{1}{w_{mb}\left(T\right)}\sum_{i=1}^{T}{\left(i-\mu_{mb}\right)}^{2}*P\left(i\right),$$
(20)$$\sigma_{bc}^{2}\left(T\right)=\frac{1}{w_{bc}\left(T\right)}\sum_{i=1}^{T}{\left(i-\mu_{bc}\right)}^{2}*P\left(i\right)$$

The between-class variance is defined as per Equation (21).
(21)$$\begin{array}{l} \sigma_{mb}^{2}=\sigma^{2}-\sigma_{w}^{2}\\ \sigma_{mb}^{2}=w_{mb}*\sigma_{mb}^{2}+w_{bc}*\sigma_{bc}^{2}+w_{mb}*{\left(\mu_{mb}-\mu \right)}^{2}+w_{bc}*{\left(\mu_{bc}-\mu \right)}^{2}\\ \sigma_{mb}^{2}=w_{mb}*w_{bc}{\left(\mu_{mb}-\mu_{bc}\right)}^{2} \end{array}$$

The procedure for determining the appropriate threshold value (T) is related to the maximum variance between groups. According to Equation (17), if $\left(W_{mb}=\sum_{i=1}^{T}P\left(i\right)=0.138\right)$ and sets the probability of pixels in the background image W_mb_ = {0.138, 0.291, 0.268} while if $\left(W_{bc}=\sum_{i=T+1}^{L}P\left(i\right)=0.638\right)$ and sets the probability of pixels in the background image W_bc_ = {0.638, 0.734, 0.725}. According to Equation (18), if $\mu_{mb}=$ {0.138, 0.121, 0.110} and $\mu_{bc}=$ {0.710, 0.821, 2.823}, the average background and breast cancer pixels can be calculated by $\mu_{mb}=\sum_{i=1}^{T}P\left(i\right)/W_{mb}$ and $\mu_{bc}=\sum_{i=T+1}^{L}P\left(i\right)/W_{bc}$, respectively. Therefore, the maximum variance of each group was calculated using Equation (21), which is $\sigma_{mb}^{2}=$ {0.028, 0.212, 1.427}. If the maximum variance between groups occurs in three iterations, then the maximum variance is 0.028 + 0.212 + 1.427 = 0.556. Hence, in this binary detection process, the optimal threshold setting is 0.556, with 1 for the breast cancer pixel and 0 for the background pixel. The highlighted breast cancer regions were white, while the highlighted mammogram background was black (see Figure 8).

#### 2.4.6. Improved Binary Image Using Mathematical Dilation Operators

Besides the binary detection applied in the previous section, morphological dilation operators are essential for more accurate detection [35]. The dilation is a procedure that enables the pixels to grow along these lines and fill the little holes, i.e., similar to increasing the thickness of the areas and associating the separate points. This operation is mainly performed post-segmenting of the characteristics of regions in mammogram images and adopts the logical processes by using the structure element involving the growing mammogram pixels in a binary image. The dilation of A represented by B is calculated in Equation (22).
(22)$$A⊕B=\left(z|{\left(\hat{B}\right)}_{z}\cap A\neq \emptyset \right)$$
where B represents the structuring element, is a reflection of set B and ∅ is the empty set. Furthermore, the detailed explanation of the method used in this paper is as follows; (1) Input a binary image by thresholding (Figure 9a) and then the breast cancer region in the image that is filled using mathematical dilation operators; (2) Assign mathematical dilation operators with structuring elements is set to 7 using a 3 × 3 mask. Dilation operation has effects of smoothing the contour of breast cancer, eliminating thin protrusions, and breaking narrow isthmuses (see Figure 9b,c) Save the resulting image after applying step 2, and the pixels of the breast cancer are preserved, and all small pixels removed after dilation by using thresholding methods [36] again.

This is an iterative algorithm for automatic estimation of threshold T given as follows: (1) An initial threshold from the image intensity histogram values T and partitions the image into two classes, G1 and G2, using the threshold, (2) Calculate the mean intensity values for G1, G2→mean(G1), and mean(G2), (3) Select a new threshold value using T = 0.55 × (mean(G1) + mean(G2)), (4) Repeat steps 1–3 until the mean intensity values G1, and G2 in successive iterations do not change ∆T. In this step, start means gray level and ∆T = 0. The algorithm results in T = 0.67 after five iterations, so T = 0.67 is used. The breast cancer region is shown in Figure 9c.

#### 2.4.7. Edge Detection

As the segmented image in Section 2.4.6, we perform some experiments to compare with various edge detectors such as the Sobel edge operator [37], Robert edge operator [38], Prewitt edge operator [39], and Canny edge operator [39], for the segmentation of the boundaries of breast cancer on Figure 9c. Figure 10 shows the result in this case. According to Figure 10a, the Sobel edge detector will be more accurate in detecting the boundaries of breast cancer and more flexible to unseen image datasets.

## 3. Results

The outcomes of the proposed method by using KM, FCM, FKM, and KM++CSO show breast cancer detection through mammogram images from three different datasets. The proposed method was evaluated by SEN, SPEC, and ACC [40] and the Jaccard index score [41]. SEN, SPEC, and ACC were used for standard quality measurement of the segmented output image. SEN is the percentage of breast cancer pixels segmented as breast cancer pixels, and SPEC is the percentage of background detected as background pixels. ACC represents the percentage of the overall pixels that are segmented correctly. These indexes are defined by Equations (23)–(25).
(23)$$Sensitivity\;(SEN)=\frac{TP}{TP+FN}*100$$
(24)$$Specificity\;(SPEC)=\frac{TN}{FP+TN}*100$$
(25)$$Accuracy\;(ACC)=\frac{TN+TN}{\left(TP+TN\right)+\left(FP+FN\right)}*100$$
where TP (True Positive) is the number of breast cancer pixels that are detected as abnormal, FN (False Negative) is the number of breast cancer pixels that are detected as normal, TN (True Negative) is the number of backgrounds that are detected as normal, and FP (False Positive) is the number of backgrounds that are detected as abnormal. Another metric to evaluate the breast cancer detection between the proposed method and manually detected with three experts was used for comparative analysis, which is widely used in the literal. Another performance aspect of the Jaccard index score [41] is defined as per Equation (26).
(26)$$Jaccard\;Index=\frac{\left|ROI_{KM++CSA}\cap ROI_{GT}\right|}{\left|ROI_{KM++CSA}\cup ROI_{GT}\right|}$$
where ROI_KM++CSA_ was the breast cancer region by using the proposed detection method, ROI_GT_ was the breast cancer region of the ground truth image and |ROI_KM++CSA_| is its cardinality.

### 3.1. Breast Cancer Detection Results

The proposed methods are implemented on MATLAB version 2020a (The MathWorks Inc.) and run on a 4.00 GHz Intel(R) Core (TM) i7–6700K CPU, 8GB (RAM) under the Microsoft Windows 10, 32–bit operating system. In Figure 11, we list images with four different detection methods and a number of clusters for every clustering method. Figure 11(a1) is the original image with signs of breast cancer. Figure 11(a2) shows the results of applying KM to detect breast cancer. Figure 11(a3) shows the effect of using OTSU’s threshold on the detected image of Figure 11(a2). Figure 11(a4) is the result of running a mathematical morphology on the image (Figure 11(a3)) and applying global thresholding. The detection results revealed that the detected images generated by morphological dilation are the best way to fill the gaps in the segmented stage. Figure 11(a5) shows the result of applying the Sobel edge operator. We tested the proposed method with some images in which SEN, SPEC, ACC, and Jaccard index score indices evaluated the method′s effectiveness. Selecting a range of values for a number of clusters k would lead to a not-so-good detection result. If the value for the number of clusters is decreased, the colors of the neighborhood may be confused. Moreover, KM requests that the user indicate the number of clusters k before the detection commences. The number of clusters the user indicates must compare with the number of gray colors. It is not essential to have prior knowledge of the number of gray colors contained by the image since there is an arrangement made for re–inputting the number of clusters as soon as KM gets to the end of the clusters, which indicates that it stops. In this experiment, the proposed method results with the test image can be seen in Figure 11(b1–b5). Figure 11(b1) is the original image of breast cancer. Figure 11(b2) is the result of applying FCM with four clusters performed using k = 4. Figure 11(b3) shows the binary detection results for Figure 11(b2) using the OTSU’s threshold. The result of applying mathematical morphology based on dilation operators and running global thresholding on the dilated image is shown in Figure 11(b4). Figure 11(b5) displays the result of applying the Sobel edge operator. FCM iterates are based on the number of clusters and come over on the considered image. Unlike KM, the FCM will return the number of clusters after the detection. Next, FKM was tested on a mammogram image, the gray level image, and their results are shown in Figure 11(c1–c5). The image shown in Figure 11(c2) displays the four clusters performed by choosing k = 4 and the detected image. Figure 11(c3) shows the result of using the OTSU’s threshold method. Figure 11(c4) shows the results of applying mathematical morphology based on dilation operators and running global thresholding on the image. Figure 11(c5) displays the result of using the Sobel edge operator. FKM could be an algorithm produced from FCM and KM but carries more of an FKM property than KM. Finally, KM++CSO is tested on mammogram images, and its performance is compared to that of other machine learning like KM, FCM, and FKM. The first test image Figure 11(d1), is displayed to compare the experimental results of breast cancer. Figure 11(d2) shows the four clusters performed using k = 4 and the detected image. Figure 11(d3) is applied to detect breast cancer regions using OTSU’s method with threshold values of 0 to 255 and set to 128. The breast cancer is dilated from its background using morphological–based dilation operators, and the results of applying global thresholding are shown in Figure 11(d4). The Sobel edge is again used to detect edges in the regions. By observing Figure 11(d5), we can say that the Sobel edge detects the edges. KM++CSO works on gray-level images like FCM and creates the same number of iterations as in KM and FKM. Time is taken to detect based on the tested image using KM++CSO to be faster than other clustering methods, whereas, in a few cases, KM also shows up to be faster than the original FCM and FKM. Whereas KM++CSO and KM compete in time, FCM has been modified to create the same number of iterations as FKM with a faster operation time. That is, FKM is faster than FCM. The conflict in time between FCM and FKM and KM and KM++CSO is expected to account for the properties of the image under consideration and the efficiency of the machine on which the algorithms are tested.

In terms of average accuracy, the number of iterations is considered. The more iterations, the more accuracy. The iteration that FCM and FKM can perform generally depends on the number of gray images contained by an image, which limits its iterative ability, unlike that of KM++CSO, which segments based on the number of clusters assigned in an image. As a result, FCM and FKM are less accurate than the KM and KM++CSO. The performance of the proposed methods on three open-access datasets is shown in Table 2. They have been tested in four unsupervised machine learnings: KM, FCM, FKM, and MK++CSO. Typically, the training dataset is generally for the samples required; at present, the methods are built with a series of image processing methods, which might still need a training set for a constant learning rate and an optimization for all parameters. From the data analysis, the FCM used on 123 images of BCDR successfully obtained the SEN, SPEC, and ACC rates of 61.29%, 61.19%, and 61.20%, respectively. However, using KM++CSO, the SEN, SPEC, and ACC rate increased to 96.90%, 96.16%, and 96.42%, respectively. Therefore, the accuracy has been increased to 35.22%. For FCM used on 123 breast cancer images of Mini–MIAS, the algorithm successfully obtained the SEN, SPEC, and ACC rates of 60.67%, 60.97%, and 60.78%. Then, KM++CSO achieved SEN, SPEC, and ACC rates of 96.78%, 97.10%, and 96.92%, respectively. Therefore, the rate of accuracy has been increased to 36.14%. Also, 2620 images have been used to evaluate and compare the performance of the fusion methods. Similarly, for DDSM, we obtain SEN, SPEC, and ACC. FCM received the SEN, SPEC, and ACC rates of 67.22%, 69.12%, and 61.10%. Then, KM++CSO achieved SEN, SPEC, and ACC rates of 95.68%, 95.10%, and 95.49%, respectively. Therefore, the accuracy has been increased to 26.39%. For this reason, the detection method has achieved perfect detection results. Overall, the proposed method demonstrated that the detection results are good quality and suitable for the breast cancer detection region from the mammogram image. However, comparing the proposed methods directly with other methods is difficult because of the difference in the number of images, differences in the dataset, and evaluation performance of detection.

### 3.2. Manual Detection Versus Measurement

In this section, the proposed detection method has been evaluated of the breast cancer region by three experts versus measurements automatically detected by the proposed methods. The correlation between the area of breast cancer detected and the measures created by three experts is calculated as per Equation (26). For BCDR, this paper used 123 images based on KM++CSO, and we obtained 112 images successfully, while 11 images were misdiagnosed. Therefore, the proposed algorithm achieved a Jaccard index score of 91.05%. For Mini–MIAS, the proposed method accurately detected 289 images out of 323. Therefore, it is shown that 289 images with a Jaccard index score of 89.47% are detected correctly, while 34 images with a score of 10.53% are detected incorrectly. Moreover, the method accurately detected 2289 images out of 2620 images (87.37% Jaccard index score) in DDSM, while 331 images with a score of 12.63% were detected incorrectly. Figure 12 illustrates the segmentation results for mammogram images randomly selected from BCDR, Mini–MIAS, and DDSM databases. The first segmentation of Figure 12a is demonstrated as the best breast cancer segmentation, which obtained a 91.05% Jaccard Index score on the BCDR database. The segmentation results of Figure 12b correctly depict the breast cancer region, which received an 89.47% Jaccard Index score on the Mini–MIAS. The segmentation results of Figure 12c correctly display the breast cancer region, which obtained an 87.37% Jaccard Index score on the DDSM. Based on the experimental results of our proposed method, its suitability for breast cancer segmentation has been successfully verified. Moreover, the proposed method outperformed breast cancer segmentation compared with BCDR, Mini–MIAS, and DDSM datasets. However, the proposed method obtained lower results by SEN, SPEC, ACC, and Jaccard Index scores in the DDSM for breast cancer segmentation.

## 4. Discussion

The diagnosis system of artificial intelligence in medicine has increased in recent years, which may be seen in the number of published articles [2,3,4,5,6,7,8,9,10,11,12,13,14]. The KM++ based on the CSO method for mammogram evaluation used within this work presented high sensitivity, specificity, and accuracy for the segmentation of breast cancer. The following contributions to the research community were provided in this article:

(1) We provided an overview of the breast cancer segmentation methods on three publicly available datasets, optimizing the parameters of KM++CSO for each dataset. With KM++CSO, the best SEN, SPEC, ACC, and Jaccard index scores in detecting breast cancer using mammogram images among the four methods are 96.90%, 96.16%, 96.42%, and 91.05%, respectively. A description of the experimental results is provided in Table 2.

(2) When it comes to the detection of healthy and breast cancer–infected mammogram images, the time taken to segment based on the tested image with a KM++CSO appears to be faster than the KM, FCM and FKM whereas, in a few cases, KM also shows up to be faster than FCM and FKM. The average execution time required for KM, FCM, and FKM to get the optimal solution with three clusters is 7, 12, and 9 seconds per image, respectively, using the computer specifications described earlier. The average time for KM++CSO for the same cluster with the same clusters and iterations is 4 seconds per image. The proposition of the new method and its performance are shown in Section 3.1.

(3) It is also shown that the pre-processing methods have improved the accuracy and enhanced the coarse detection performance of the KM++CSO, including OTSU’s thresholding and mathematical morphology. This paper indicates that by cutting–edge performance in KM++ and applying CSO, it is possible to achieve a higher level of performance, improving breast cancer detection accuracy.

(4) A new method for accurately detecting breast cancer was proposed based on KM++CSO in mammogram images. The technique allows for more accurate detection of breast cancer by tracking the lesions on the mammogram surface. It can also aid the experts in the manual classification and detection of breast cancer. The proposed methods can produce good detection with no effort in terms of coarse to fine detection of breast cancer in a mammogram image. 

The results from the mammogram image analysis revealed the possibility of the successful detection of breast cancer. The results from the image segmentation confirmed differences in the position, shape, and cancer features between the healthy image and people with breast cancer. However, there are limitations to this work. The proposed method is evaluated on small datasets, although it provides data for appropriate segmentation analysis. The second limitation is setting the multiple algorithms to compare with the ground truth image as the basis of three expert reports. This process requires more complexity in time. However, for future work, the execution time in the detection process is a region of interest. Sometimes, the methodology, which does not require fixing the number of clusters before detecting, may be used for coarse detection.

## 5. Conclusions

Within this work’s limitations, we can conclude that the tested KM++CSO can be helpful for an initial evaluation of breast cancer screening for medical diagnostics. Moreover, the result of the proposed method outperformed breast cancer segmentation when compared with the existing approach on BCDR, Mini–MIAS, and DDSM datasets.

## Figures and Tables

**Figure 1 diagnostics-12-03088-f001:**
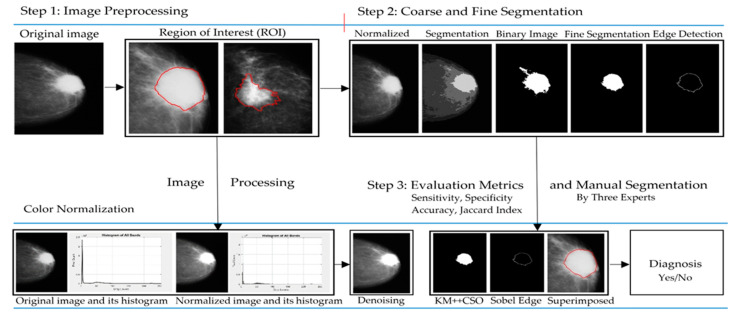
The proposed KM++CSO for segmentation of breast cancer region from mammogram image. The red line represents the area of breast cancer labeled by the specialist.

**Figure 2 diagnostics-12-03088-f002:**
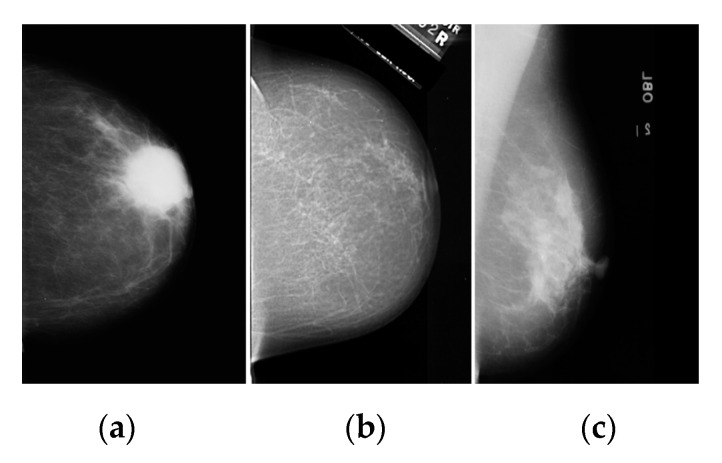
The original image from three open-access datasets: (**a**) BCDR, (**b**) DDSM, (**c**) Mini–MIAS.

**Figure 3 diagnostics-12-03088-f003:**
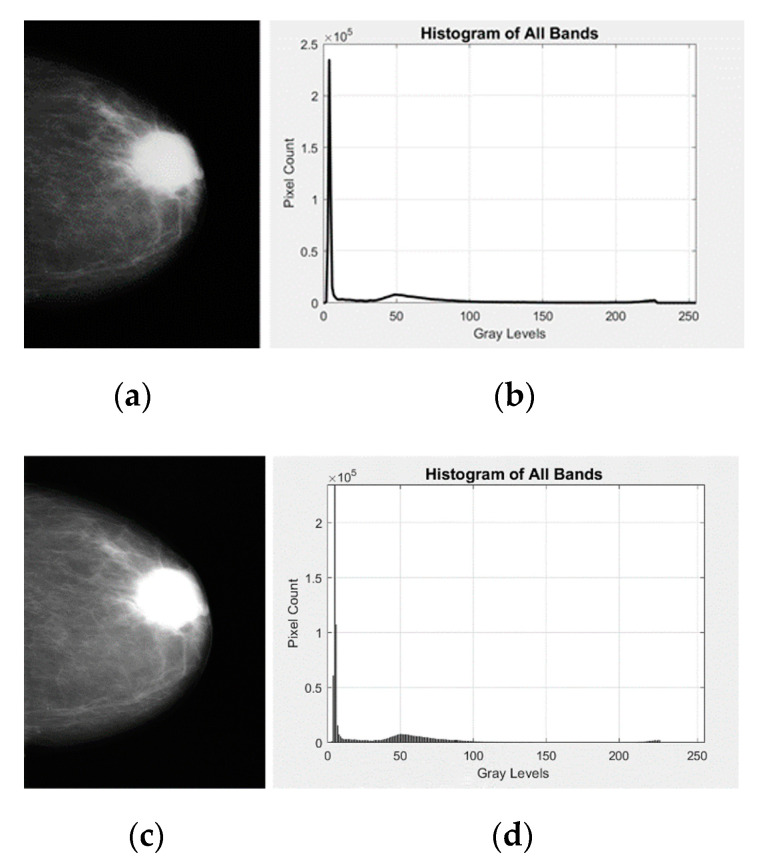
The mammogram image and it is after histogram specification with their respective histograms: (**a**) looking at the mammogram image, in which the original image appears to contain mostly the same grayscale value, (**b**) histogram of the original mammogram image, (**c**) the subtle changes in grayscale images become for the mammogram image to classify in the normalized image, and (**d**) the peak in the normalized image is approximately 50 grayscale levels, as seen in the normalized histogram. Fortunately, the process improves the segmentation of breast cancer in mammogram images, and it simultaneously enhances the noise in mammogram images.

**Figure 4 diagnostics-12-03088-f004:**
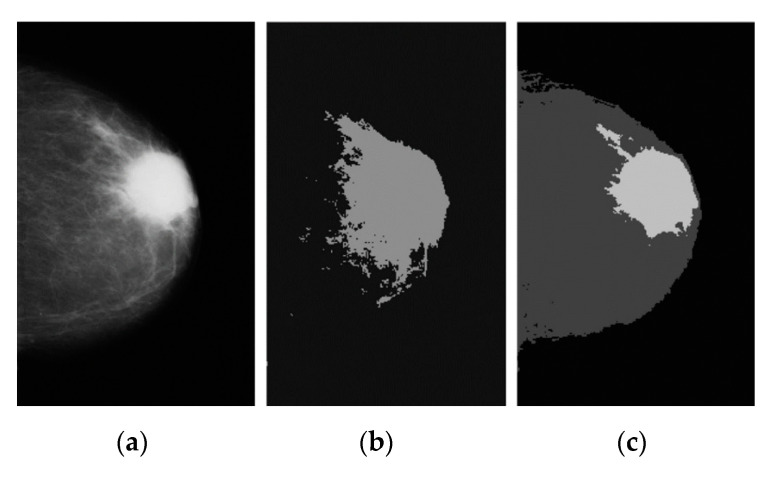
An original mammogram image is shown in a KM segmented version: (**a**) pre-processed image, (**b**) the first image segmented with k = 2, (**c**) good quality segmented with k = 3, (**d**) another example of segmenting breast cancer with k = 4, (**e**) segmenting cancer with k = 5, and (**f**) region segmented as candidate cancer area with k = 6 after 110 iterations.

**Figure 5 diagnostics-12-03088-f005:**
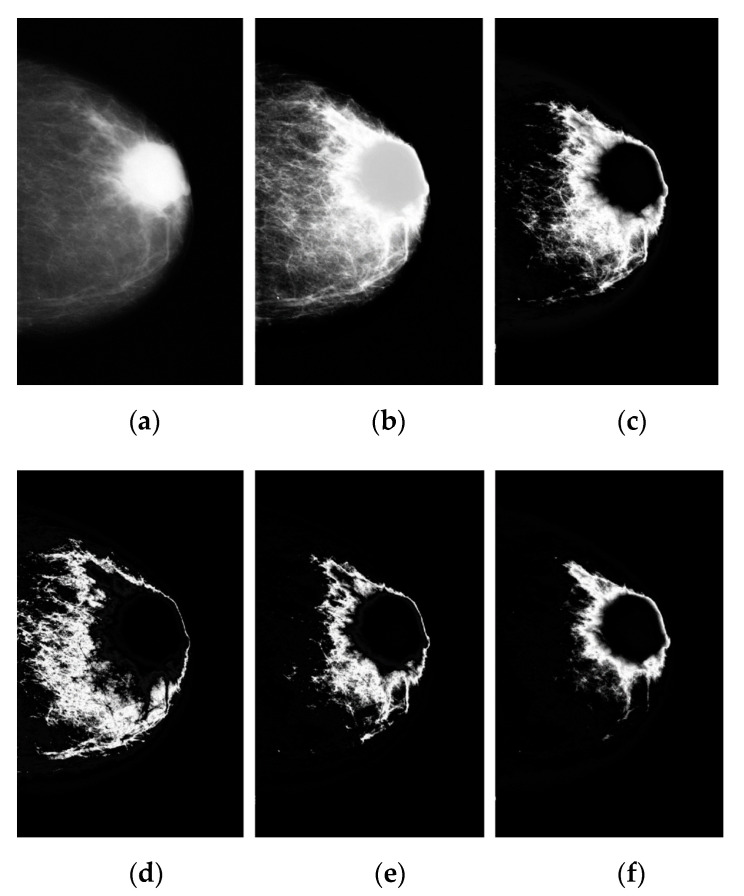
The segmentation of breast cancer using the FCM method: (**a**) preprocessed image on the left, (**b**–**f**) results of breast cancer regions segmented with k = 2, 3, 4, 5, and 6 after 100 iterations.

**Figure 6 diagnostics-12-03088-f006:**
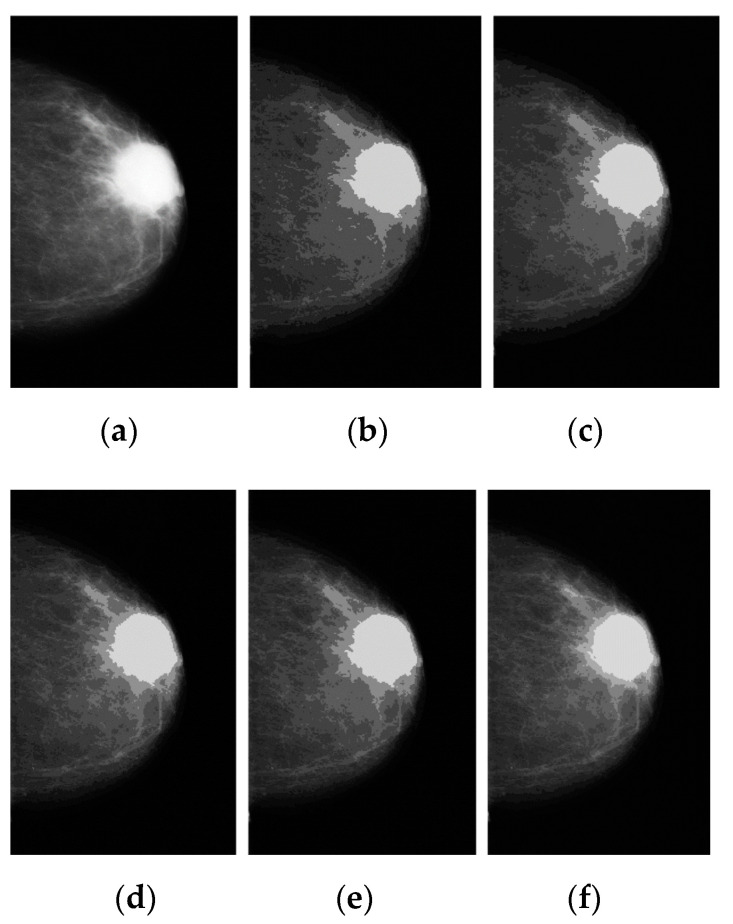
The segmentation of breast cancer using the FKM method: (**a**) preprocessed image on the left, (**b**–**f**) results of breast cancer segmentation on the gray level image with k = 2, 3, 4, 5, and 6 after 100 iterations.

**Figure 7 diagnostics-12-03088-f007:**
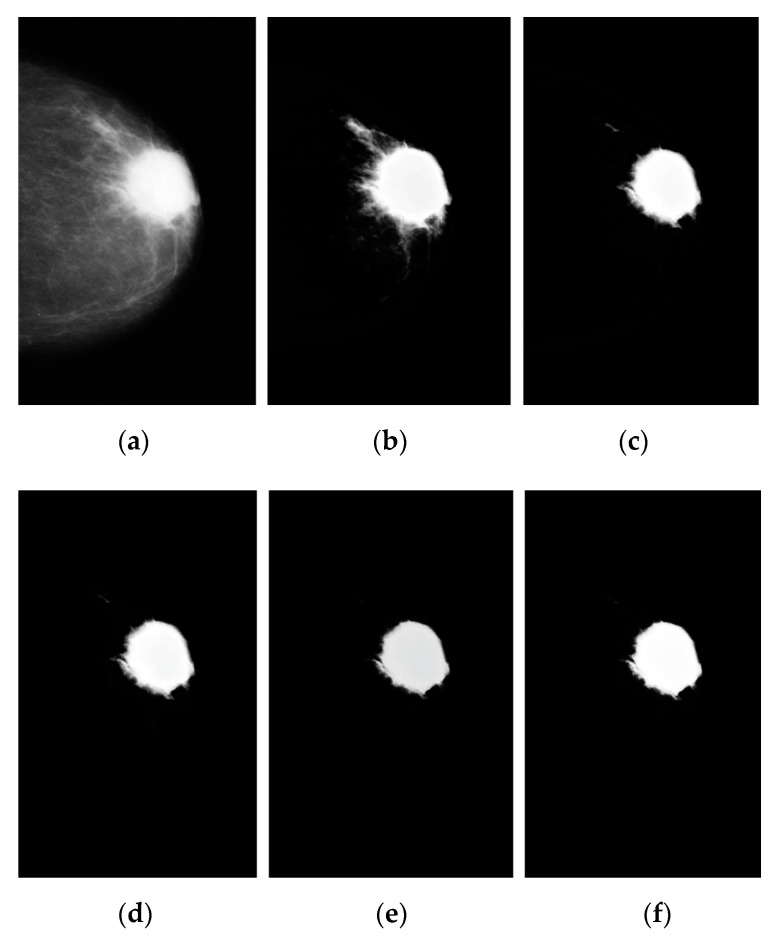
The segmentation of breast cancer using KM++CSO, (**a**) preprocessed image on the left, (**b**–**f**) the results of breast cancer segmentation with k = 2, 3, 4, 5, and 6 after 110 iterations.

**Figure 8 diagnostics-12-03088-f008:**
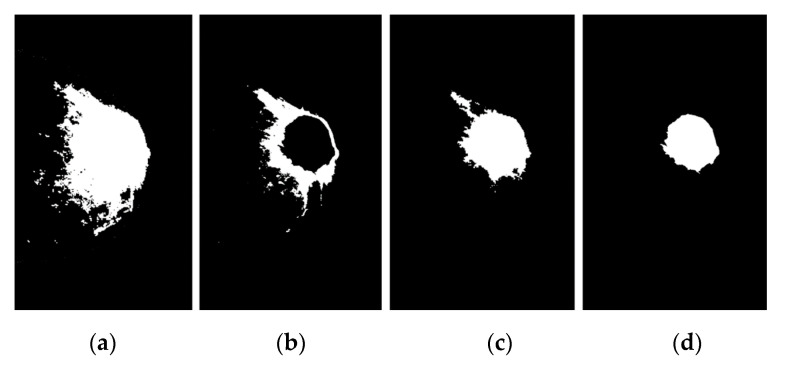
Breast cancer image and results of four segmentation methods, then, obtained binary image based on OTSU’s threshold with T = 0.556 gives the best result, (**a**) KM method, (**b**) FCM method, (**c**) FKM method, (**d**) KM++CSO method.

**Figure 9 diagnostics-12-03088-f009:**
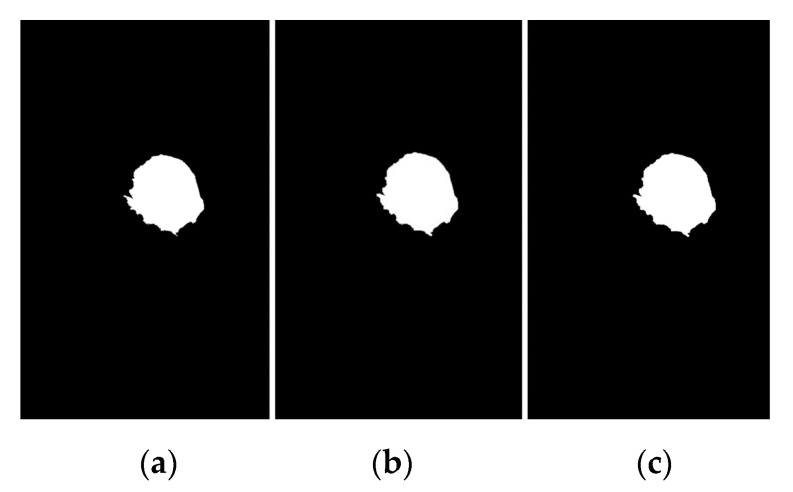
Segmenting and removing small pixels by morphological modeling based on a dilation operator, (**a**) breast cancer segmented using KM++CSO and OTSU’s threshold method, (**b**) image reconstructed with structure element of 7 to disconnect component pixels of the binary image, (**c**) regional maxima image preserved and all small objects removed using global thresholding.

**Figure 10 diagnostics-12-03088-f010:**
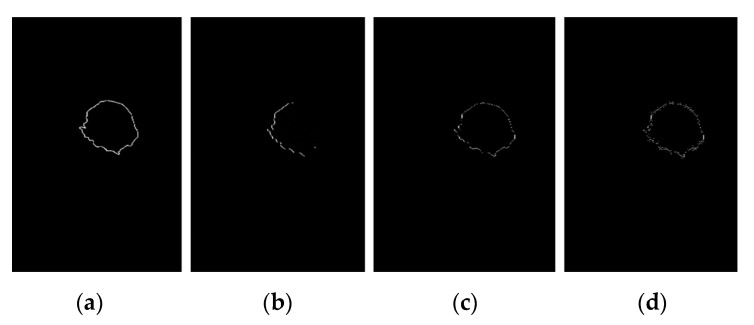
The mammogram image is segmented with various edge operator, (**a**) Sobel edge operator, (**b**) Robert edge operator, (**c**) Prewitt edge operator, (**d**) Canny edge operator. Canny edge segmentation results are different from the other methods. Prewitt and Robert’s edge operators produce almost the same edge segmentation. However, the Sobel edge segmentation result is superior by far to the other methods.

**Figure 11 diagnostics-12-03088-f011:**
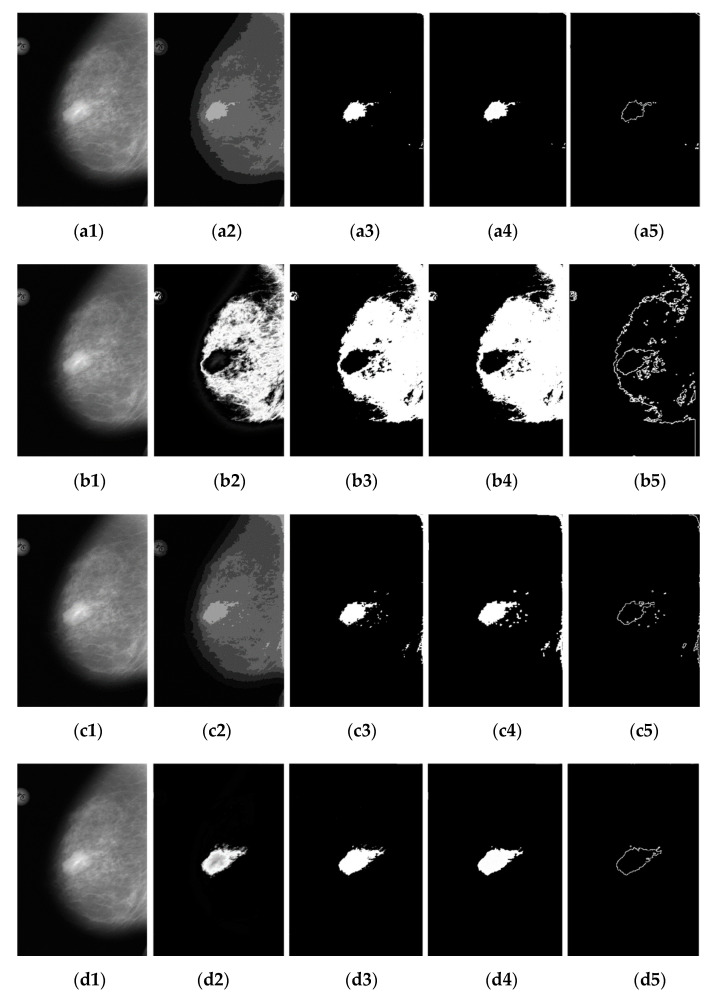
Brest cancer image and result of four detection method with OTSU’s threshold, mathematical modeling and Sobel edge operator, (**a1**–**a5**) KM, (**b1**–**b5**) FCM, (**c1**–**c5**) FKM, (**d1**–**d5**) KM++CSO.

**Figure 12 diagnostics-12-03088-f012:**
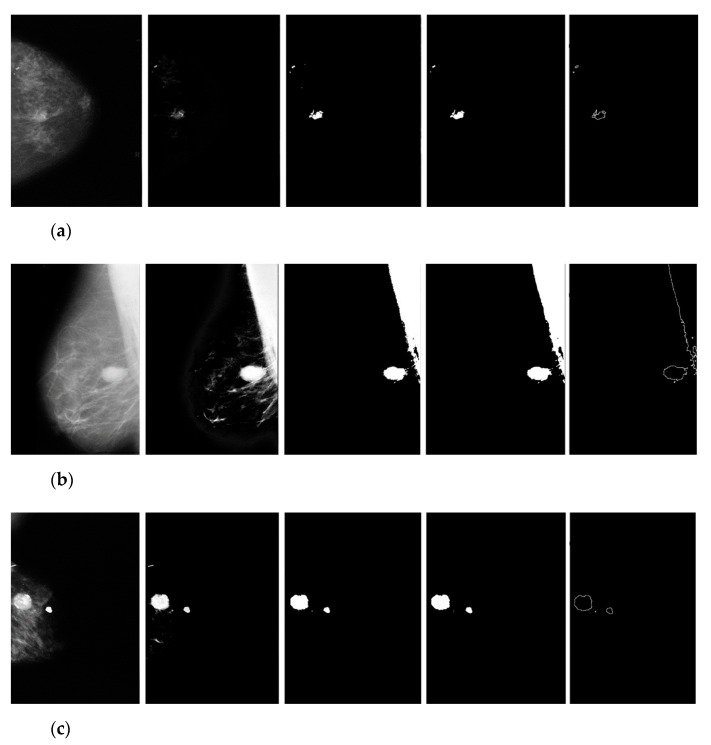
Examples of results of breast cancer segmentation from different databases: (**a**) BCDR, (**b**) Mini–MIAS, (**c**) DDSM.

**Table 1 diagnostics-12-03088-t001:** Comparison of machine learning and deep learning methods for segmentation of breast cancer.

Authors	Methods	Dataset	SEN (%)	SPEC (%)	ACC (%)
Karabatak et al. [2]	Naïve Bayesian	Wisconsin breast cancer	99.11	98.25	98.54
Pak et al. [3]	SVM and ANN	Mini–MIAS and DDSM	SVM = 100	SVM = 80	SVM = 92.85
			ANN = 87.50	ANN = 93.33	ANN = 91.31
Yang et al. [5]	PSCSM	Mini–MIAS	–	–	90.9
Punita et al. [6]	RGM, MRF segmentation, Stochastic Relaxation methods, and Fuzzy	DDSM	98.10	97.8	98.00
Al-masni et al. [7]	Regional deep learning based on CNN and CLAHE	IN Breast	97.14	92.41	95.64
Shen et al. [8]	MSGDM	IN Breast	97.56	88.89	94.16
Punita et al. [9]	Fuzzy Rough Segmentation (FR), Rough Segmentation (RS), and Fuzzy Segmentation (FS)	Mini–MIAS	FR = 91.23	FR = 91.23	FR = 97.45
			RS = 80.25	RS = 79.45	RS = 82.45
			FS = 89.89	FS = 89.12	FS = 90.12
Zebari et al. [10]	The thresholding method and binary morphological for segmentation	Mini–MIAS, IN BCDR	98.9	98.4	98.13
Anji et al. [11]	DNNS	MG Cancer	–	–	97.21
Azlan et al. [12]	PCA and SVM	DDSM and Mini–MIAS	93.94	96.61	95.24
George et al. [13]	The hybrid methods by combined the CNN and SVM	BreaKHis	97.24	96.18	96.21
Alanazi et al. [14]	CNN architectures	Kaggle 162 H&E	-	-	87.00
Mohammed et al. [15]	SVM	Mini–MIAS	-	-	91.12
Agnes et al. [16]	MA-CNN	Mini–MIAS	96.00	-	-
Yektaei et al. [17]	Multiscale Convolutional Neural Network (MCNN)	Mini–MIAS	95.90	-	97.03
Kaur et al. [18]	Multiclass Support Vector Machine (MSVM)	Mini–MIAS	-	-	96.90
Viswanath et al. [19]	SVM	Raw sample images	–	68	84.84
Jin et al. [20]	Binary classifier with CNNI–BCC	Mini–MIAS	–	–	73.24
Kayode et al. [21]	SVM	Mini–MIAS	94.4	91.3	–
Debelee et al. [22]	SVM and MLP	Mini–MIAS	SVM = 99.48	SVM = 98.16	SVM = 99
			MLP = 97.40	MLP = 96.26	MLP = 97
Zhang et al. [23]	Mask R–CNN	DCE–MRI	80	74	75

**Table 2 diagnostics-12-03088-t002:** Comparison between the proposed segmentation method in terms of SEN, SPEC, ACC, and Jaccard index score.

Methods	Datasets	Percentage			
		SEN (%)	SPEC (%)	ACC (%)	Jaccard Index (%)
KM	DDSM	88.38	89.10	88.89	80.45 (2108/2620)
FCM		67.22	69.12	69.10	70.95 (1859/2620)
FKM		90.15	90.18	90.16	83.55 (2189/2620)
KM++CSO		95.68	95.10	95.49	87.37 (2289/2620)
KM	Mini–MIAS	90.10	89.78	89.92	85.13 (275/323)
FCM		60.67	60.97	60.78	64.08 (207/323)
FKM		91.12	91.20	91.14	86.99 (281/323)
KM++CSO		96.78	97.10	96.92	89.47 (289/323)
KM	BCDR	89.96	89.89	89.92	78.86 (97/123)
FCM		61.29	61.19	61.20	65.85 (81/123)
FKM		91.40	91.56	91.48	88.62 (109/123)
KM++CSO		96.90	96.16	96.42	91.05 (112/123)

## Data Availability

Not applicable.

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
