# Peer review of "Breast Cancer Detection in Mammogram Images Using K–Means++ Clustering Based on Cuckoo Search Optimization"

_diagnostics, 2022, doi:10.3390/diagnostics12123088_

Round 1
Reviewer 1 Report
I feel a graphicl abstract would make it easier for the readers to grasp the conclusion of your work
The manuscript is well-written and well-presented
The title has to be simpler
Author Response
Response Letter
Comment |
Response |
Point 1: I feel a graphical abstract would make it easier for the readers to grasp the conclusion of your work |
We have added details of the graphical abstract, as shown in Fig. 1. |
Point 2: The manuscript is well-written and well-presented |
Thank you so much, Prof. |
Point 3: The title has to be simpler |
The title reflects the study’s data image, method, and introduction. |
Thank you very much for your kind consideration. We look forward to hearing from you soon.
Warmest regards,
Assoc. Prof. Dr Kittipol Wisaeng, Ph.D.
Electrical and Computer Engineering
Reviewer 2 Report
"This is an interesting research manuscript regarding breast cancer detection in mammogram images by Kittipol Wisaeng. The author describes some methods of solving old problems regarding the accuracy of various programs in detecting breast cancer. In my view, the used equations are correct but need to be validated by competent people.
Minor comments:
- Abbreviations should be explained at first appearance in the abstract section and the main text of the manuscript
- Lines 12-13 - “The pre-processing method is used to improve the proposed KM++CSO method more segmentation efficiently” – not sure about the meaning of this idea
- The Introduction section should be rewritten to avoid repeating words and syntaxes (especially type “Kalpana et al.”, “Ariana et al.” and so on)
- Line 130 – “Some authors, such as [31],…” please rewrite
- Line 133 – “Instead, [32] proposed stain normalization..” – please rewrite
- Eq (2), (3) and (4) - I think that there are some overlapping characters
177, 179, and so on – “we chose”, “we combine”- you and who else?
Major comment:
- Considering the importance of such a study in the face of alarming mortality and morbidity caused by breast cancer, maybe the author can propose an approachable and practical design for applying these machine learning methods in daily clinical practice. Because the usefulness of such methods is uncontested, but the practical approach is usually harder to be implemented."
Author Response
Response Letter
Comment |
Response |
Point 1: Abbreviations should be explained at first appearance in the abstract section and the main text of the manuscript |
Revised |
Point 2: The pre-processing method is used to improve the proposed KM++CSO method more segmentation efficiently” – not sure about the meaning of this idea. |
The original image is usually low quality. Therefore, the pre-processing step is important to feed the enhanced image to the segmentation process, thus giving better results than using the original image. |
Point 3: The Introduction section should be rewritten to avoid repeating words and syntaxes (especially type “Kalpana et al.”, “Ariana et al.” and so on) |
We have revised and rewritten based on your feedback. |
Point 4: Line 130 – “Some authors, such as [31],…” please rewrite |
Bejnordi et al. [31], |
Point 5: Line 133 – “Instead, [32] proposed stain normalization..” – please rewrite |
Janowczyk et al. [32] |
Point 6: Eq (2), (3) and (4) - I think that there are some overlapping characters |
Not same Prof. |
Point 7: Considering the importance of such a study in the face of alarming mortality and morbidity caused by breast cancer, maybe the author can propose an approachable and practical design for applying these machine learning methods in daily clinical practice. Because the usefulness of such methods is uncontested, but the practical approach is usually harder to be implemented." |
Revised in main text with yellow highlighted. |
Thank you very much for your kind consideration. We look forward to hearing from you soon.
Warmest regards,
Assoc. Prof. Dr Kittipol Wisaeng
Round 2
Reviewer 2 Report
I believe that the current version of the manuscript adds value to the information presented.
The answers provided are sufficient.
I have no further comments.
Congratulations on the idea and good luck!